# Cyclodehydrogenation of molecular nanographene precursors catalyzed by atomic hydrogen

Rafal Zuzak [1], Pawel Dabczynski[1], Jesús Castro-Esteban [2,6], José Ignacio Martínez [3], Mads Engelund [4], Dolores Pérez [2], Diego Peña [2,5] & Szymon Godlewski [1]

Atomically precise synthesis of graphene nanostructures on semiconductors and insulators has been a formidable challenge. In particular, the metallic substrates needed to catalyze cyclodehydrogenative planarization reactions limit subsequent applications that exploit the electronic and/or magnetic structure of graphene derivatives. Here, we introduce a protocol in which an on-surface reaction is initiated and carried out regardless of the substrate type. We demonstrate that, counterintuitively, atomic hydrogen can play the role of a catalyst in the cyclodehydrogenative planarization reaction. The high efficiency of the method is demonstrated by the nanographene synthesis on metallic Au, semiconducting $TiO_2$, Ge:H, as well as on inert and insulating $Si/SiO_2$ and thin NaCl layers. The hydrogen-catalyzed cyclodehydrogenation reaction reported here leads towards the integration of graphene derivatives in optoelectronic devices as well as developing the field of on-surface synthesis by means of catalytic transformations. It also inspires merging of atomically shaped graphene-based nanostructures with low-dimensional inorganic units into functional devices.

The diverse structural shapes of nanographenes[1–8] and graphene nanoribbons (GNRs)[9–13] can exhibit electronic[14] and magnetic[12,13] properties that arise from their distinctive electron-confinement geometries. Their synthesis has been approached by using solution organic chemistry to deliver molecular precursors that undergo surface-catalyzed inter- and intramolecular transformations under ultrahigh vacuum (UHV) conditions after deposition on the target substrate surface[15]. For example, planarization of a precursor through cyclodehydrogenative C-C coupling yields polycyclic, conjugated graphene-based structures[9]. This crucial step can be reliably and efficiently initiated on noble metal surfaces[9,15]. However, the high density of metallic substrate electronic states also affects the electronic properties of the product molecule and devices fabricated on such substrates and has prompted a search for alternative routes[16]. Examples of on-surface synthesis on nonmetallic substrates are rarely reported[16–28], and the planarization pathways have been limited to individual examples of surface-assisted dehydrogenation[29], cyclodehydrogenation[30] or cyclodehydrofluorination[31,32] on very specific faces of rutile titania crystal.

We now report a strategy that allows the formation of planar $sp^2$ carbon-based moieties on a range of substrates. Atomic hydrogen has been used for on-surface synthesis experiments to remove

[1]Faculty of Physics, Astronomy and Applied Computer Science, Jagiellonian University, PL 30-348 Krakow, Poland. [2]Centro de Investigación en Química Biolóxica e Materiais Moleculares (CiQUS) and Departamento de Química Orgánica, Universidade de Santiago de Compostela, 15782 Santiago de Compostela, Spain. [3]Instituto de Ciencia de Materiales de Madrid, Consejo Superior de Investigaciones Científicas (ICMM-CSIC), 28049 Madrid, Spain. [4]Espeem S.A.R.L. (espeem.com), L-4206 Esch-sur-Alzette, Luxembourg. [5]Oportunius, Galician Innovation Agency (GAIN), 15702 Santiago de Compostela, Spain. [6]Present address: Department of Chemistry, Massachusetts Institute of Technology, 77 Massachusetts Avenue, Cambridge, MA 02139, USA. ✉e-mail: diego.pena@usc.es; szymon.godlewski@uj.edu.pl

polymerization by-products[33,34], protect reactive edges[35], induce heteroatom[33] or isotope substitution[36], tailor reactions[37], quench organometallic states[38], hydrogenate GNRs[39] or induce intermolecular fusion[40]. Recently the role of atomic hydrogen in H shift and elimination steps has also been investigated[41]. We report dosing of atomic hydrogen to initiate the cyclodehydrogenation reaction of precursors in a manner independent of the substrate (Fig. 1). The approach was verified based on high-resolution scanning tunneling microscopy (STM) imaging, bond-resolved non-contact atomic force microscopy (nc-AFM) visualization[42] and time-of-flight secondary-ion mass spectrometry (ToF-SIMS) measurements with metallic, semiconducting, and insulating substrates. Climbing-image nudged-elastic band (CI-NEB) calculations were used to explore the reaction path theoretically.

To test this approach, we selected 10,10'-dibromo-9,9'-bianthracene (DBBA) as a precursor of GNRs 1, and specially designed and synthesized compounds 5 and 6, as molecular precursors of nanographenes 2 and 3, respectively. These precursors allowed us to analyze the planarization of both polymeric units in 4, as well as structures containing phenyl substituents with free rotation (5) and helical moieties whose geometrical orientation is governed by the steric hindrance (6).

## Results and discussion
### Cyclodehydrogenation initiated by atomic hydrogen
We started the experiments with the Au(111) surface, which serves as an example of the metallic substrate, and also allowed for the in-depth analysis of the reaction products by both high-resolution STM and bond-resolved nc-AFM microscopy[42]. Figure 2a shows the typical STM image of 7-armchair graphene nanoribbons (7-AGNRs, 1) synthesized on Au(111) by dosing of atomic hydrogen. The GNRs are fabricated in a two-step process: (i) DBBA was polymerized thermally at 200 °C by means of Ullmann-like coupling to obtain polyanthryl 4, and (ii) these polymers were subjected to atomic hydrogen dosed for 30 min at $1\cdot10^{-7}$ mbar (molecular hydrogen gas pressure) while keeping the sample at 220 °C.

The STM topography showed the presence of mainly defect-free 7-AGNRs (1); the two blue arrows indicate the occurrence of incompletely planarized units (Fig. 2a). The planarization was very efficient, and in general more than 99% of additional new C-C bonds are formed and associated benzene rings between anthracene units are already created, leaving the non-fully reacted unfinished GNR structures indicated by arrows as sole examples (for details see Supplementary Fig. 3). During the hydrogen dosing procedure bromine atoms, which

are the polymerization reaction by-products, were removed from the surface[33] as HBr molecules[43].

In our experiment, the sample temperature during dosing was selected to ensure efficiency of the cyclodehydrogenation process (see Supplementary Fig. 3 for details). The generation of 7-AGNRs presented here was achieved at 220 °C, well below the temperature range (>320 °C) at which the catalytic activity of Au(111) promotes cyclodehydrogenation[9,15]. This lower temperature suggested that a different reaction pathway was initiated by dosing with atomic hydrogen, which could be examined by closer examination of the GNR structures. In Fig. 2b, we distinguished different terminations at both ends of the GNR, which is caused by the attachment of two hydrogen atoms at the center of one termini and formation of a $sp^3$ carbon methylene moiety (-CH₂-), as visualized by bond-resolved nc-AFM images supplemented by high-resolution STM topographies in Fig. 2c–f. The difference in appearance of the two termini in high-resolution STM pictures was caused by the methylene unit, which quenched the radical character associated with the zig-zag termination[44,45].

This presence of either perfect zig-zag or methylene-bridged (previously named as superhydrogenated[40]) edges enabled unambiguous identification of the reaction products. Crucially, the nc-AFM images of arm-chair sides of the GNRs (Fig. 2d, f) did not show the features associated with formation of methylene moieties and had perfect flat-edge termination. The STM appearance of superhydrogenated zig-zag edges (Fig. 2b, e) is in accordance with ref. 45. and suggest that unwanted additional hydrogen atoms were removed after annealing at 300 °C[35]. In our experiment, we estimate the abundance of superhydrogenated zig-zag edges at ≈66% (158 out of 240 counted zig-zag termini), and that after annealing to 300 °C, the fraction of methylene units was reduced to <5% leaving almost exclusively perfectly shaped and terminated 7-AGNRs based on $sp^2$ carbon atoms.

To demonstrate the versatility of the approach, we used hydrogen-assisted planarization to form nanographenes 2. The bond-resolved nc-AFM (Fig. 2g, inset) shows the atomic structure of 2 generated with atomic hydrogen at 220 °C. We note that the synthesis of nanoflakes 2 by purely thermal means cannot be achieved <300 °C, and that the perfect shaping of the edges indicates that no additional hydrogen atoms (methylene moieties) were present. Additionally, in the temperature range of our hydrogenation experiments (up to 220 °C), we did not observe any signs of intermolecular cross-coupling as reported for molecules activated with higher pressure of atomic hydrogen[40].

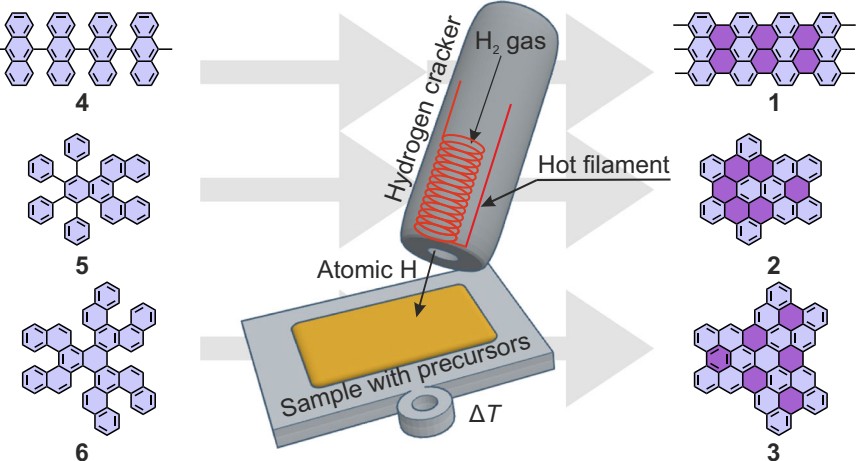

**Fig. 1 | Cyclodehydrogenation catalyzed by atomic hydrogen.** Polymers 4 and flexible precursors 5, 6 are transformed into graphene derivatives 1, 2, 3 through hydrogen catalyzed cyclodehydrogenation. Light violet marks hexagonal rings present in molecular precursors 4, 5, 6, whereas dark violet in 1, 2, 3 indicates rings generated through hydrogen catalyzed cyclodehydrogenation.

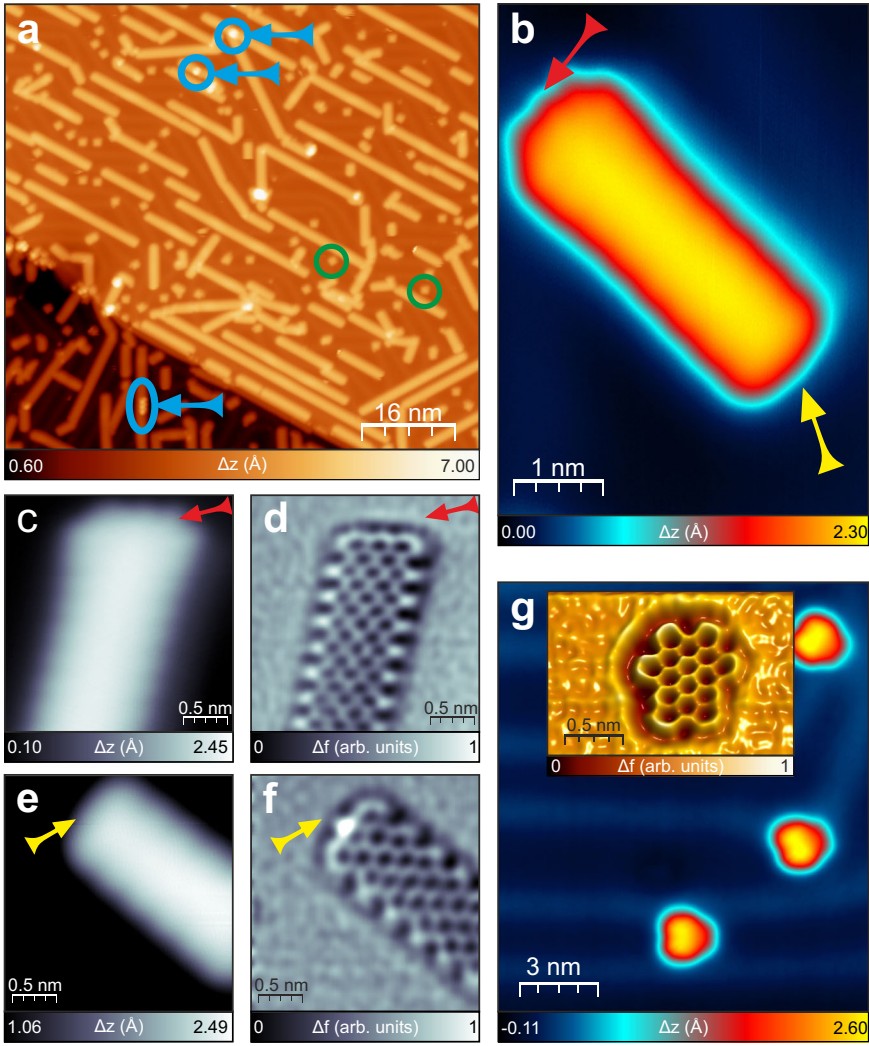

**Fig. 2 | Cyclodehydrogenation promoted by atomic hydrogen at 220 °C on Au(111). a** Large-scale STM image with 7-AGNRs (**1**). Blue arrows and circles/ovals indicate rarely observed incompletely planarized DBBA units, green circles mark single nanographenes formed through planarization of DBBA. **b** Filled state STM image of the 7-AGNR (**1**) with differently terminated zig-zag edges. High-resolution STM (**c**) and bond-resolved AFM (**d**) images of perfectly shaped 7-AGNR (**1**) with zig-zag termini indicated by red arrows. STM (**e**) and nc-AFM (**f**) images of 7-AGNR (**1**) with methylene (*sp*³ carbon) bridged end indicated by yellow arrows. **g** High-resolution STM image of nanographenes **2**. Inset shows bond-resolved AFM image visualizing perfect shaping of **2**. Tunneling current: 50 pA (**a**), 25 pA (**b–g**), bias voltage: -1 V. Source data are provided as a Source Data file.

We replaced the metallic substrate with a semiconductor surface, $TiO_2(110) - (1 \times 1)$, which allowed us to compare directly with the recently reported thermally initiated cyclodehydrogenation of the same precursors that occurs at 400 °C[30]. We analyzed the ability to efficiently initiate cyclodehydrogenation with atomic hydrogen both between vicinal phenyl rings rotating around $\sigma$ bonds in **5** as well as within strained pentahelicenes of **6**. The generated target nanographenes **2** and **3**, respectively, are shown in Fig. 3. Compounds **2** and **3** could be discerned in the high-resolution STM images acquired at voltages corresponding to the frontier orbitals (for **2** HOMO: −1.15 V, LUMO: +2.25 V; for **3** HOMO: -1.45 V, LUMO: +2.45 V)[30] visualized in Fig. 3a, f.

To unambiguously demonstrate planarization of both precursors **5** and **6**, STM imaging with voltages adjusted to the energy levels of **2** and **3** molecular states were acquired following the spectroscopic (STS) characterization of **2** and **3** in ref. 29. Comparison with image simulations in Fig. 3b–e and Fig. 3g–j, confirmed the synthesis of **2** and **3** with atomic hydrogen.

The efficient cyclodehydrogenation initiated under similar conditions on both Au(111) and TiO₂(110)-(1×1) suggested that, in our

experiments, the externally dosed atomic hydrogen, not the substrate, catalyzed the reaction. This may seem counterintuitive given that atomic hydrogen dosing facilitated the removal of covalently-bonded hydrogen. Figure 3k shows a mechanistic proposal for the conversion of **6** into **3**, which is inspired by Sánchez-Sánchez and co-workers reporting on-surface hydrogen-induced covalent coupling of poly-cyclic aromatic hydrocarbons[40]. Atomic hydrogen likely added to compound **6** to form various $\pi$ radicals. If this radical was placed in the internal region (fjord region) of a pentahelicene moiety (for example, **7**), it would continue to react to form an intramolecular C-C bond, followed by a sequence of C-H cleavage reactions to obtain structure **8**. Similar transformations in the other internal regions (fjord regions) of the intermediate molecules would lead to the flat nanographene **3**.

To validate the mechanistic proposal (Fig. 3k) we have performed a comprehensive series of theoretical atomistic simulations using pentahelicene as a proof-of-concept model system for the cyclode-hydrogenation reaction. The pentahelicene unit could be regarded as a representative of the molecular internal region (fjord region) involved in the sequential cyclodehydrogenative planarization of compound **6**. The calculated mechanism progression involves: (i) atomic H addition

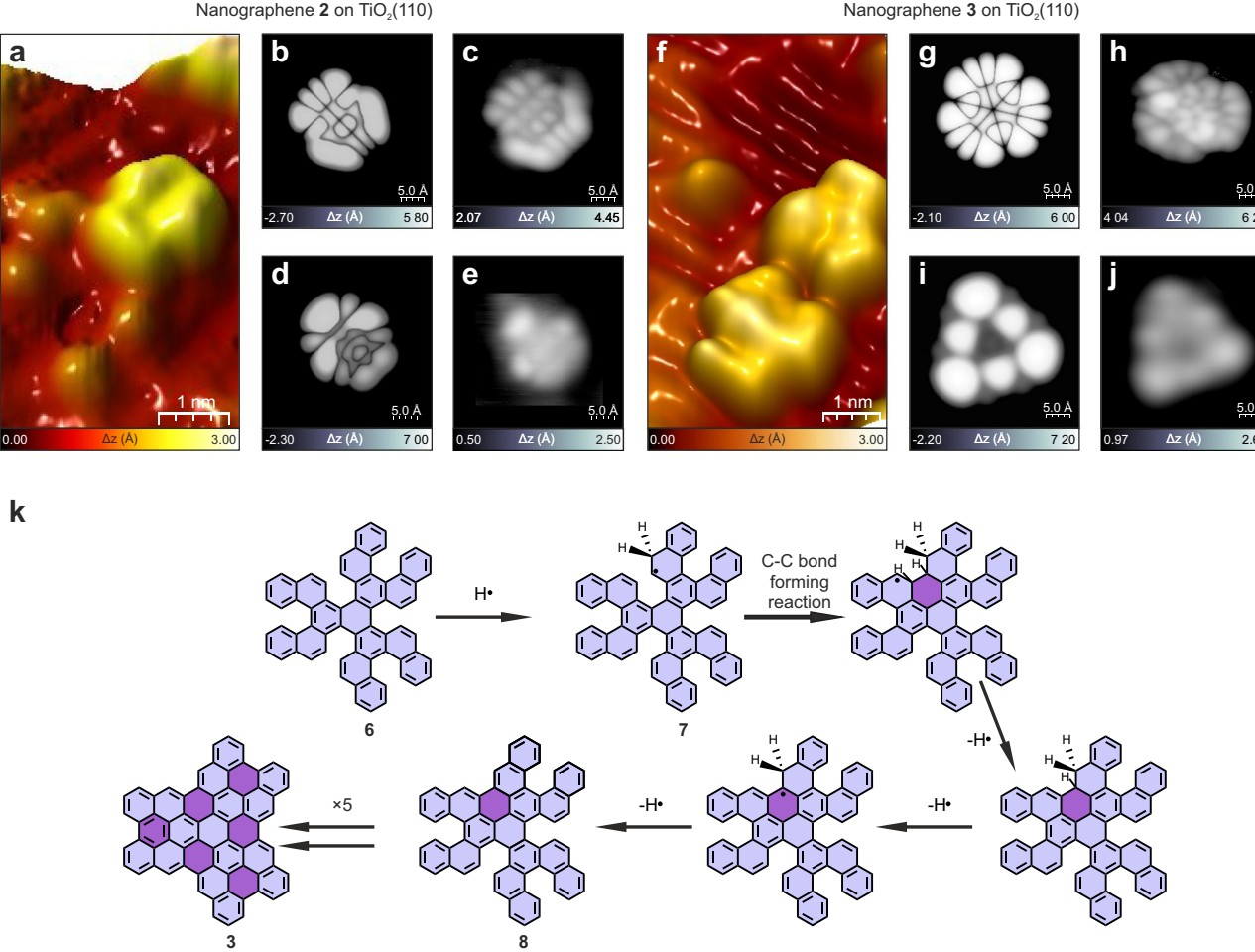

Fig. 3 | **Atomic hydrogen induced planarization yielding nanographenes 2 and 3 on TiO$_2$(110)-(1 × 1). a** 3D STM view of nanographenes **2**. High-resolution simulated (**b**) and experimental (**c**) filled-state images of **2**. High-resolution simulated (**d**) and experimental (**e**) empty state images of **2**. **f** 3D STM image of nanographene **3**. High-resolution simulated (**g**) and experimental (**h**) filled state images of **3**. High-resolution simulated (**i**) and experimental (**j**) empty state images of **3**. **k** Plausible reaction mechanism for planarization of **6** into **3** catalyzed by atomic hydrogen. Light violet marks hexagonal rings present in molecular precursor **6**, whereas dark violet in intermediates and nanographene **3** indicates rings generated through hydrogen catalyzed cyclodehydrogenation. Bias voltage: +1.5 V (**a** –**f**), −1.7 V (**c**), +2.3 V (**e**), −1.5 V (**h**), +2.2 V (**j**), tunneling current: 10 pA (**c**, **j**), 15 pA (**a**–**h**). Source data are provided as a Source Data file.

to the peripheral C atom forming a π-radical (energy barrier: 0.08 eV), which activates the molecule for (ii) cyclization through C-C coupling generating a six-membered ring (energy barrier: 0.56 eV). Subsequently, we propose (iii) three consecutive Eley-Rideal hydrogen abstraction reactions[46] forming three gas-phase H$_2$ molecules (maximum energy barrier of 0.01 eV), which lead to the full planarization of the molecular structure. All intermediate sub-reactions exhibit net free energy gains ranging from -0.5 to -3.3 eV. Therefore, the aforementioned C-C bond formation is the limiting reaction step, featuring a moderate energy barrier of 0.56 eV making the process highly probable in the experimentally applied temperature range in typical times of around $10^{-4}$ s (for details see Supplementary Fig. 4). These gas-phase calculations ensure that limiting reaction step energy barrier of 0.56 eV as a true upper bound for the on-surface cases. Influence of any of the substrates under study in the atomistic mechanism would just decrease that energy barrier value due to different factors like, e.g.: (i) a higher available amount of atomic hydrogen diffusing on the surface (increasing the atomic hydrogen capture cross-section by the deposited precursors, boosting the planarization process in time and efficiency), (ii) the appearance of interfacial image potentials (case of Au), yielding Coulomb-like attracting image forces, or (iii) the surface-driven electrostatic interaction leading to slight structural distortions of the precursors inducing the appearance of intrinsic molecular

dipoles, thus enhancing the catalysis of the process. Besides, additional calculations reveal that, although very unlikely, a simultaneous double H addition, rather than a single H addition to one peripheral C atom, but to both peripheral C atoms (each near the coupling C-C atoms leading to cyclization), eliminates the cyclization energy barrier.

**Efficient planarization on semiconductors by atomic hydrogen**
The above-described route should enable the generation of nanoflakes and GNRs on surfaces that do not catalyze these reactions. To demonstrate this possibility, we used TiO$_2$(011)-(2 × 1), which promotes Ullmann-like polymerization[17,18] but cannot be used for thermally-driven cyclodehydrogenation[31,32]. Figure 4 demonstrates the successful synthesis of 7-AGNRs (**1**) from standard DBBA precursors through a two-step process: (i) intermolecular thermally driven (260 °C) polymerization yielding DBBA-based polymers **4**[17] (Fig. 4a) followed by (ii) hydrogen-initiated cyclodehydrogenation (atomic hydrogen, 220 °C) to afford 7-AGNRs (Fig. 4b). The planarization was supported by the apparent height comparison of the polymers and GNRs (green and red curves in upper panel of Fig. 4c, respectively), as well as by the STS characterization of the electronic end states associated with zig-zag termini of GNRs displayed in Fig. 4c (lower panel). The STS results were in good agreement with previous reports for 7-AGNRs generated

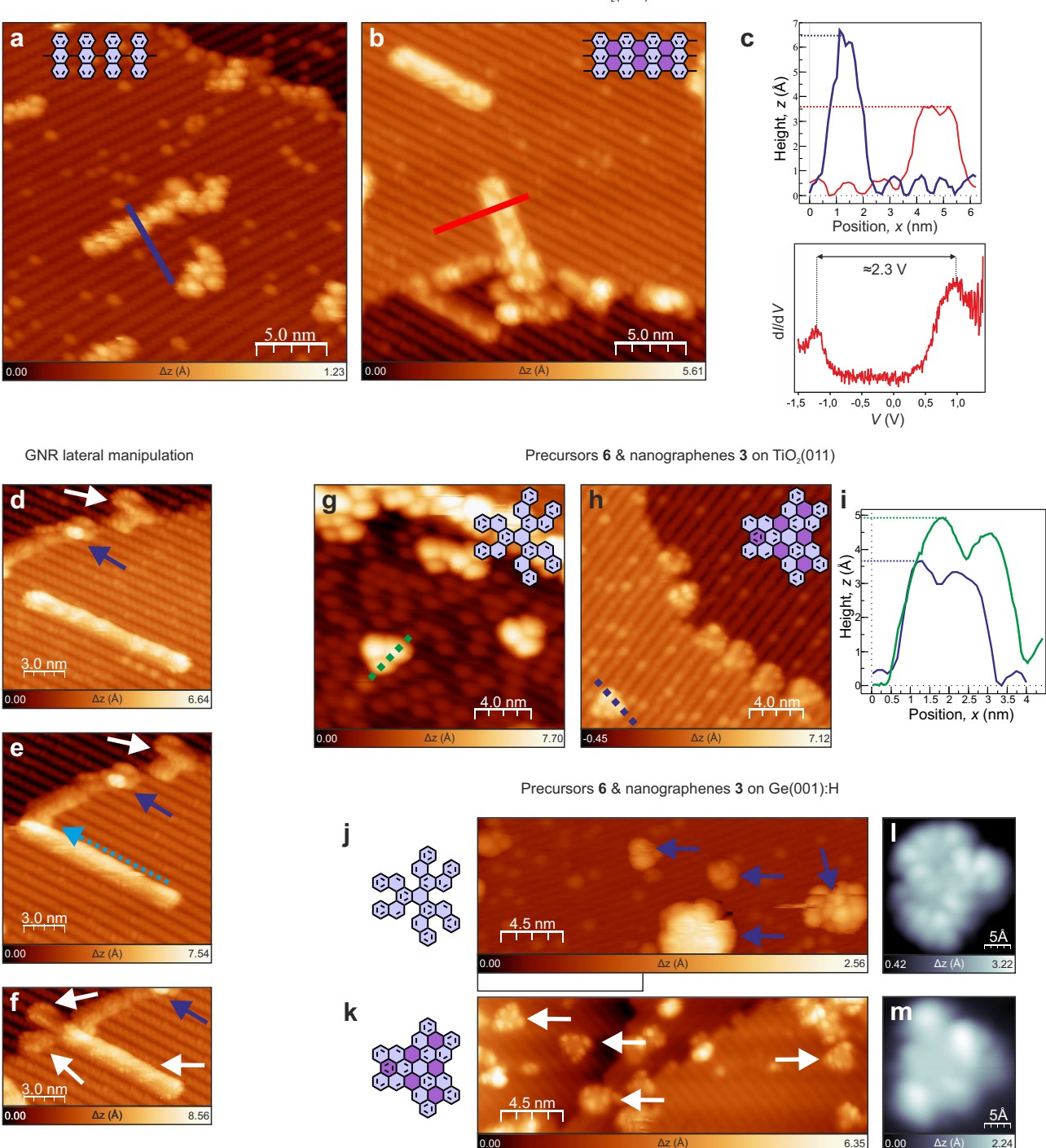

**Fig. 4 | Synthesis of 7-AGNRs on TiO₂(011)-(2×1) and nanographenes 3 on TiO₂(011)-(2×1) and Ge(001):H catalyzed by atomic hydrogen.** Typical STM images of DBBA based polymers **4** (**a**) and 7-AGNRs **1** (**b**) after atomic hydrogen treatment. Upper panel in (**c**) shows profiles acquired along violet and red lines in (**a**) and (**b**), respectively, and indicates easily discernible apparent height difference between DBBA-based polymers and 7-AGNRs. Lower panel in (**c**) shows occurrence of electronic states associated with zig-zag end of 7-AGNR. **d** Long 7-AGNR oriented along the surface reconstruction rows. **e** Lateral manipulation of 7-AGNR. Light blue arrow indicates direction of GNR movement, dark blue arrow indicates a defect playing a role of a position marker. **f** 7-AGNR manipulated over the surface step edge. White arrows indicate clearly discernible characteristic appearance of the GNR termini. STM image of precursors **6** (**g**) and nanographenes **3** (**h**). **i** Profiles acquired along green and blue dashed lines in (**g**) and (**h**), respectively. Profiles indicate the noticeable difference of the apparent height of **6** and **3**. Molecular precursors **6** (**j**) and nanographenes **3** (**k**) on Ge(001):H. Dark blue arrows indicate precursors **6**, whereas white arrows nanographenes **3**. STM images of **3** on Ge(001):H acquired at voltages corresponding to HOMO (**l**) and LUMO (**m**). Light violet in molecular schemes marks hexagonal rings present in molecular precursors **4**, **6**, whereas dark violet in **1**, **3** indicates rings generated through hydrogen catalyzed cyclodehydrogenation. Tunneling current: 5 pA (**a**), 7 pA (**h**), 10 pA (**f**–**k**), 15 pA (**b**–**g**), 20 pA (**j**), 100 pA (**d**, **e**), bias voltage: +1.5 V (**a** through **f**), +2.3 V (**g**), −1.5 V (**h**, **j**), +2.2 V (**i**, **k**). Source data are provided as a Source Data file.

through the cyclodehydrodefluorination from specially designed molecular precursors[32].

Furthermore, our approach delivered not only GNRs immobilized by surface steps or domain boundaries, but also mobile ones that usually aligned with the reconstruction rows of the substrate. An example of a mobile GNR is presented in Fig. 4d supplemented by the demonstration of the tip-induced manipulation (Fig. 4e) that led to the GNRs displacement over the surface step edge imaged in Fig. 4f. The ends of the GNR were clearly identified by the characteristic features originating from the distinct electronic states associated with the zig-zag edge[32]. The demonstration of the successful planarization induced by atomic hydrogen (220 °C) was further extended to the synthesis of nanographenes 3 from nonplanar starting material 6, as shown by high-resolution STM images in Fig. 4g–i, as well as the closed layer of 3 (Supplementary Figs. 5, 6). Additionally, we demonstrate the synthesis of 3 on semiconducting, passivated Ge(001):H surface (Fig. 4j, k) accompanied by STM orbital imaging (Fig. 4l, m). Importantly, nanographenes 3 could not be at all synthesized on both $TiO_2(011)\text{-}(2\times1)$ and Ge(001):H by thermal treatment without the application of atomic hydrogen.

## Nanographenes synthesized directly on insulators

Synthesizing graphene materials on insulating surfaces is more challenging because of the lower interaction energy between the precursor and the substrate compared to metals and semiconductors, which may promote desorption rather than cyclodehydrogenation. In addition, the atomic-scale characterization of reaction products is also challenging because STM cannot be used on insulating surfaces. The hydrogen-induced planarization reaches high efficiency between 200 and 220 °C, and at these temperature internal cyclodehydrogenation may compete with desorption. Indeed, we observed the thermal desorption of precursor 5. However, precursor 6 showed stronger interaction with the substrate because of its size and shape, which facilitated the formation (through cyclodehydrogenation) of the planarized nanographene 3.

We used the silicon covered with ≈300 nm of silicon dioxide ($SiO_2$) as a substrate to demonstrate on-surface synthesis of nanographenes on insulators (Fig. 5a). This interface is widely applied in electronics industry[47], and it has also attracted attention as a basic substrate for fabrication of low-dimensional devices[48,49]. Importantly, the top $SiO_2$ layer is not only resistant to atomic hydrogen, but the exposure at 200 to 300 °C may even improve surface quality[50]. To demonstrate the versatility of the approach we have performed additionally similar experiments yielding nanoflakes 3 on two additional insulators: bulk NaCl (Supplementary Figs. 8, 9, 10) and thin salt layer on copper NaCl/Cu(111) (Supplementary Fig. 7).

To monitor the transformation from precursor 6 into planar nanographene 3, we applied ToF-SIMS[36,51,52]. Figure 5b presents the mass spectra for 6 deposited on $Si/SiO_2$ plotted in dark blue that showed the presence of precursors 6 by a series of peaks at 828 – 830 g mol⁻¹ that reflected the natural abundance of carbon isotopes. Additional peaks recorded at masses starting from 816 up to 827 g mol⁻¹ could be attributed to the precursors in various states of dehydrogenation induced by ToF-SIMS measurements, as already reported for precursors on Au(111)[51]. The tail extended exactly to the fully dehydrogenated nanographene 3 in full accordance with ref. 51. on a metallic Au(111) substrate.

The blue curve represents data acquired for the precursors 6 at $Si/SiO_2$ annealed to 190 °C. ToF-SIMS experiments yielded similar results and demonstrated that annealing alone was insufficient to induce transformation from 6 into target compound 3. In contrast, the green plotted curve shows data acquired for the sample with 6 after treatment with atomic hydrogen at 190 °C with clearly discernible nanographenes 3 represented by the series of peaks starting at the mass of 816 g mol⁻¹, thus demonstrating the achieved

cyclodehydrogenation on an insulating sample. However, the presence of peaks for masses corresponding to the precursors indicate the presence of unreacted 6 on the surface.

To understand the interaction of 3 and 6 on $Si/SiO_2$, we have performed static ToF-SIMS measurements as a function of the substrate temperature (Fig. 5c). For temperatures >210 °C, the unreacted precursor 6 was almost completely desorbed, whereas generated nanoflakes 3 were still adsorbed on the surface. These nanoflakes 3 were synthesized from 6 by atomic hydrogen produced during ToF-SIMS measurements (for details see Supplementary Fig. 2). In Fig. 5d, we show the ToF-SIMS data integrated up to 210 °C that illustrate the presence of precursors 6. Figure 5e shows the ToF-SIMS signal integrated from 210 to 400 °C, which indicates the presence of almost exclusively nanoflakes 3 > 210 °C.

These results suggested how to prepare a $Si/SiO_2$ surface covered only with nanoflakes 3. We prepared the $Si/SiO_2$ with precursors 6, which were then treated with atomic hydrogen flux at 190 °C, and then annealed at 240 °C to remove unreacted precursors. The resulting ToF-SIMS data are shown in Fig. 5b on top by a black curve that indicated the presence of nanoflakes 3 with strongly suppressed peaks for the precursor 6. In the experiments with $Si/SiO_2$ we took advantage of the robustness of both precursors 6 and nanoflakes 3 and transported the samples from the preparation chamber to the SIMS setup in ambient conditions. Controlled experiments with the sample transported entirely in UHV conditions showed comparable results.

To sum up, we have demonstrated that the planarization of nanographene molecular precursors could be achieved through cyclodehydrogenation catalyzed by atomic hydrogen on different substrates. This constitutes a promising synthetic route to overcome the long-standing challenge to initiate molecular transformations on inactive surfaces. Therefore, the atomically precise formation of nanographenes is no longer limited to catalytically active surfaces, providing an approach which enables performing on-surface synthesis on chemically inert surfaces.

Transferring the role of the catalyst for the planarization of graphene-like structures from a substrate to the dosage of atomic hydrogen expands the possibilities for constructing nanoscale structures from organic precursors. In particular, the proposed approach may be applied in the development of bulk and low dimensional functional devices combined with graphene-like moieties. In a wider perspective, our research showing the non-intuitive use of atomic hydrogen for cyclodehydrogenation could be extended to explore the possibility of creating non-benzenoid and doped rings, which introduce the desired modifications of electronic properties in the synthesized nanostructures. This approach opens up a design space for molecular electronics by avoiding the limitations posed by electronically disruptive substrates for catalyzing the desired molecular synthesis.

## Methods
### Sample preparation
Rutile $TiO_2$: Both rutile single crystals $TiO_2(110)\text{-}(1\times1)$ and $TiO_2(011)\text{-}(2\times1)$ were purchased from MaTecK GmbH company. The $TiO_2(110)\text{-}(1\times1)$ sample surface was prepared by repeated sputtering by Ar⁺ ions for 15 min at room temperature, followed by 10 min of AC current annealing at 780 °C. The same procedure was used for preparation of the $TiO_2(011)\text{-}(2\times1)$ surface, but the annealing temperature was set at slightly lower value of 760 °C. An infrared pyrometer was used to measure the sample temperature during the preparation process ($\varepsilon = 36\%$). After preparation STM imaging was used to check the surface quality.

Gold. Monocrystalline Au(111) surface was prepared by standard method with annealing (to 450 °C) and simultaneous Ar⁺ ion sputtering. The Au(111) monocrystalline was delivered by the SPL company. After preparation quality of the surface was checked by STM imaging.

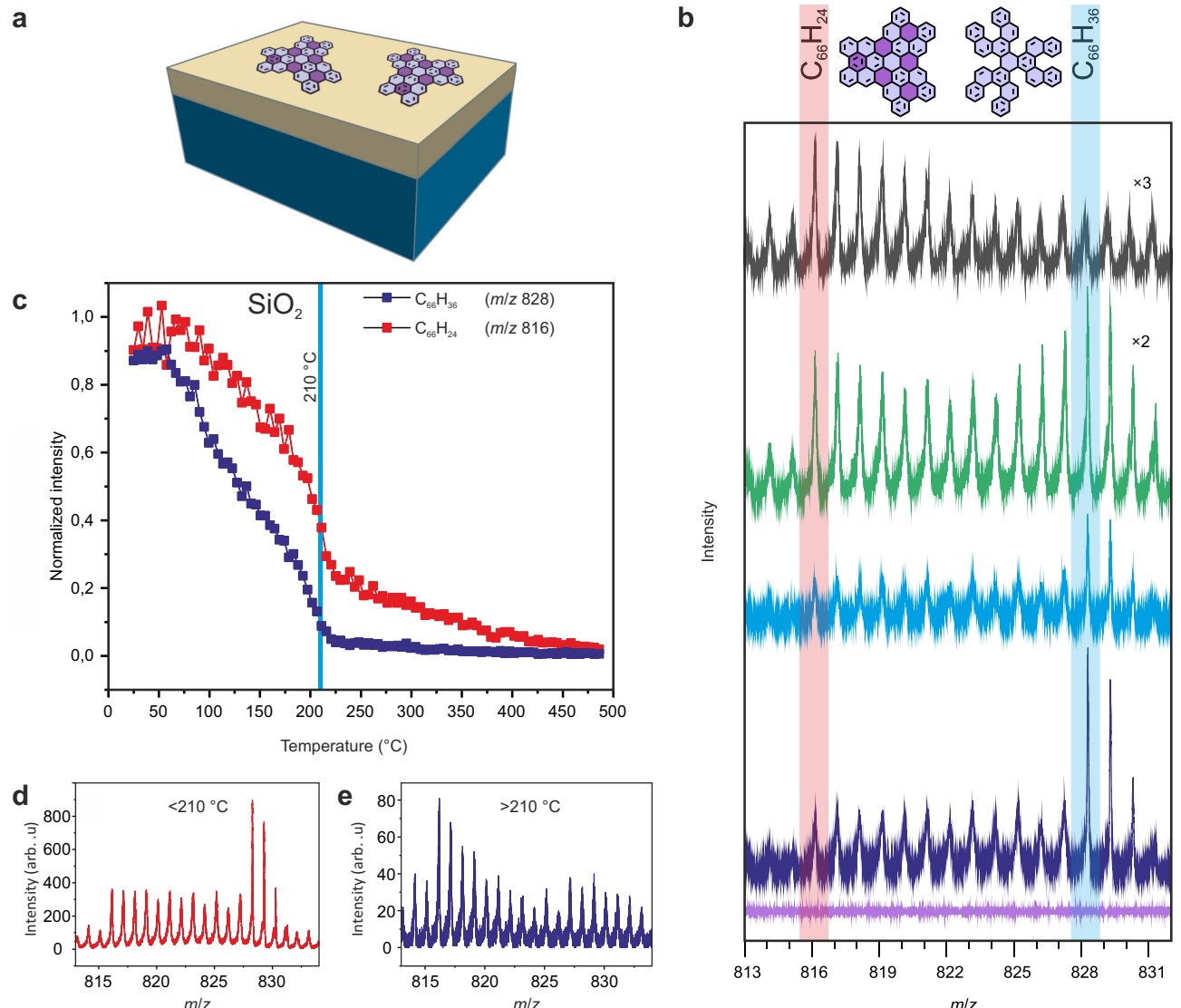

**Fig. 5 | Synthesis of nanographenes 3 on Si/SiO₂ interface induced by atomic hydrogen. a** Schematic view of nanographenes **3** on Si/SiO₂. **b** ToF-SIMS spectra with $C_{66}H_{24}$ (nanographene **3**) and $C_{66}H_{36}$ (precursor **6**) mass regions marked by transparent red and blue, respectively, colored spectra: violet – reference obtained for clean Si/SiO₂, dark blue – precursors **6** deposited on Si/SiO₂, blue – precursors **6** on Si/SiO₂ after annealing at 190 °C, green – mixture of precursors **6** and nanographenes **3** on Si/SiO₂, after annealing at 190 °C with atomic hydrogen, black – nanographenes **3** present on the surface after atomic hydrogen treatment at 190 °C and subsequent annealing at 240 °C. **c** Static ToF-SIMS performed on a Si/SiO₂ sample containing precursors **6** showing $C_{66}H_{36}$ (precursor **6**, mass 828.286 g mol⁻¹ – violet) and $C_{66}H_{24}$ (mass 816.134 g mol⁻¹ – red) signals displayed as a function of temperature. **d** Mass spectra from (**c**) integrated from room temperature up to 210 °C showing presence of precursors **6**. **e** Mass spectra from (**c**) integrated from 210 up to 400 °C indicating presence of nanoflakes **3**. In (**c**) data point – to – point normalized to the reference experiment conducted with the same primary ion dose density at room temperature. Light violet in molecular schemes marks hexagonal rings present in molecular precursors **6** ($C_{66}H_{36}$), whereas dark violet in **3** ($C_{66}H_{24}$) indicates rings generated through hydrogen catalyzed cyclodehydrogenation. Source data are provided as a Source Data file.

Ge(001):H: Ge(001) crystal was prepared by standard method with annealing (to 770 °C, direct heating) and simultaneous Ar⁺ ion sputtering. After preparation, the Ge(001) surface has been hydrogenated with a home-built hydrogen cracker (Supplementary Fig. 1). During hydrogenation temperature of the sample was set to 200 °C, and the molecular hydrogen pressure was $1 \times 10^{-7}$ mbar. After preparation quality of the surface was checked by STM imaging. The Ge(001) crystal was purchased from MaTec company.

Si/SiO₂: Si/SiO₂ samples were cut from 4-inch wafers delivered by Pi-Kem company. The samples were cleaned in alcohol (purity 99%) and acetone (purity 99%) prior to the insertion into the UHV system. After that the samples were annealed to 400 °C for 1 h (base pressure before the end of annealing at $2 \times 10^{-10}$ mbar).

NaCl (001): The Bulk NaCl sample was cleaved in the air from a bigger crystal and after transferring to the UHV system it was annealed to 350 °C for 1 h (base pressure before the end of annealing at $2 \times 10^{-10}$ mbar). The quality of the surface was checked by nc-AFM measurements.

NaCl/Cu(111): Monocrystalline Cu(111) surface was prepared by standard method with annealing (to 500 °C) and simultaneous Ar⁺ ion sputtering. The NaCl thin film on Cu(111) was prepared by evaporation of NaCl from the Knudsen cell on the Cu(111) sample kept at 200 °C to achieve the uniform 2–4 ML salt coverage.

**STM/nc-AFM measurements**

STM/nc-AFM experiment on Au and NaCl/Cu(111) was conducted at LHe temperature (≈4.5 K). All experiments for rutile TiO₂ samples were

performed at LN2 temperature (77 K). As probes we used electrochemically etched Pt-Ir STM tips and qPlus sensors (delivered by Scienta Omicron) with $Q_{factor} \approx 25{,}000$. For $dI/dV$ spectra we used the lock-in technique with the oscillation amplitude of approximately 15 mV and the oscillation frequency of 680 Hz.

To determine the influence of ambient conditions in the Si/SiO$_2$ system, initially the sample was transferred from the LT-STM machine to the ToF-SIMS UHV system both in ambient conditions and in a home-built vacuum suitcase (base pressure $\approx 5 \times 10^{-10}$ mbar). Since no measurable differences were detected for further experiments the Si/SiO$_2$ samples were transported in ambient conditions.

The NaCl bulk crystal was transferred with the applications of the vacuum suitcase, since transferring in ambient conditions proved to significantly influence the quality of the recorded ToF-SIMS data.

All STM/nc-AFM experiments were done in UHV conditions (base pressure $< 2 \times 10^{-10}$ mbar).

## Molecular material evaporation

Evaporation of molecular precursors was achieved from the water-cooled Knudsen cell manufactured by Kentax GmbH. The temperature of deposition was calibrated with the use of a quartz microbalance and was the following: 160 °C for DBBA, 190 °C for **5**, and 292 °C for **6**. The flux for all precursors was set to approximately 1 Hz per 5 min, which corresponds to the range of $3 \cdot 10^9 - 1 \cdot 10^{10}$ molecules s$^{-1}$ cm$^{-2}$ reaching the substrate surface[53].

## Hydrogen cracker

In our experiment, we used a home-built hydrogen cracker with a tungsten filament ($\Phi = 0.125$ mm). The filament was warmed up to approximately 1500–2000 °C. The H$_2$ gas was fed to the filament by a ceramic tube. The whole filament is placed in the water-cooled tube, to prevent the surrounding components from heating up. The estimated efficiency of molecular hydrogen cracking is below 10%.

Before the experiments, the cracker was outgassed to provide good base pressure in the chamber (during the experiment base pressure was at the level of $2 \times 10^{-10}$ mbar). Our standard experimental parameters for cracking are $t = 30$ min, pH$_2 = 1 \times 10^{-7}$ mbar. The disassembled cracker is shown in Supplementary Fig. 1.

## ToF-SIMS measurements

The ToF–SIMS experiments were performed with an IONTOF ToF-SIMS V (Münster, Germany) instrument, equipped with a bismuth–manganese liquid metal ion source. Static SIMS measurements were carried out with the use of Bi$^+$ 30 keV ion beam with primary ion dose density $5 \cdot 10^{11}$ ions cm$^{-2}$ to ensure static condition sSIMS. Each sample was analyzed at four randomly chosen $500 \times 500$ μm$^2$ areas. For thermal analysis the sample temperature was linearly ramped from 25 up to 395 °C at a rate of $\beta \approx 3.75$ °C min$^{-1}$. For thermal experiments the reference measurements were performed at room temperature, with the same primary ion beam dose density to determine ion beam induced damage of the sample. Mass calibration was performed with the H, H$_2$, CH, C$_2$H$_2$, C$_4$H$_4$, C$_5$H$_5$ and C$_6$H$_6$.

Supplementary Figure 2 shows estimation of the ToF-SIMS induced cyclodehydrogenation of precursors **6** into nanoflakes **3**. This is approximated by corrected intensity of 816 g mol$^{-1}$ including consideration of ToF-SIMS induced dehydrogenation[51,52]. This approach assumes that the signal at 816 g mol$^{-1}$ is the sum of the presence of nanographene **3** (generated by the atomic H produced by ToF-SIMS measurements) and ToF-SIMS induced dehydrogenation. On this basis the intensity of nanographene **3** as a function of temperature

was calculated by Eq. (1).

$$I_{\text{nanographene } 3}(T) = I_{816}(T) - I_{828}(T) * \frac{I_{816}^{\text{reference}}}{I_{828}^{\text{reference}}} \tag{1}$$

## Reaction pathway theoretical analysis

Ab-initio calculations have been carried out within the DFT framework, adequately combined with the Gibbs free energy formalism to account for the temperature effect. Localized basis set Gaussian16 atomistic simulation package[54] was employed to perform geometry optimizations and vibrational frequency analyses for all the precursor, intermediate and product structures. CI-NEB approach[55], available in the plane-wave simulation package QUANTUM ESPRESSO[56], was adopted to compute minimum energy paths (MEPs) and transition state energy barriers for all sub-reactions involved in the cyclodehydrogenative planarization reaction mechanism. Detailed explanation is available in Supp. Note 4.

## Reporting summary

Further information on research design is available in the Nature Portfolio Reporting Summary linked to this article.

## Data availability

The data that support the findings of this study are available from the corresponding authors upon request. Source data are provided with this paper.

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

## Acknowledgements

We acknowledge Dr. Łukasz Bodek for building the vacuum transfer system. We thank the following organizations for support: National Science Center, Poland, grant no. 2019/35/B/ST5/02666 (SG), Ministerio de Ciencia e Innovación, Spain, MCIN/AEI/10.13039/501100011033, grants no.: PID2022-140845OB-C62 (DiP), PID2022-139933NB-I00 (DoP), TED2021-132388B-C42 (DiP), PID2023-149077OB-C31 (JIM) and TED2021-129416A-I00 (JIM), European Regional Development Fund; Xunta de Galicia (Centro de investigación do Sistema universitario de Galicia accreditation 2023-2027), grant no. ED431G 2023/03 (DiP, DoP), European Union through ERC SyG MolDAM, grant no. 951519 (DiP). RZ was supported by NAWA—Polish National Agency for Academic Exchange, grant number BPN/BEK/2023/1/00134. The open-access publication of this article was funded by the Priority Research

Area SciMat under the program "Excellence Initiative—Research University" at the Jagiellonian University in Krakow (SG). The study was carried out using research infrastructure purchased with the funds of the European Union in the framework of the Smart Growth Operational Programme, Measure 4.2; Grant No. POIR.04.02.00-00-D001/20, "ATOMIN 2.0 - ATOMic scale science for the INnovative economy" (SG).

## Author contributions

R.Z. and S.G. conceived the project with support from Di.P. J.C.E. carried out the precursor synthesis and analysis under supervision of Do.P. and Di.P. R.Z. conducted the on-surface synthesis experiments and STM measurements with support from S.G. P.D. performed ToF-SIMS measurements and analyzed the results. M.E. prepared the theoretical simulations of STM images. J.I.M. prepared the calculations of the cyclodehydrogenation reaction pathway with support from S.G. and Di.P. S.G. and Di.P. prepared the manuscript with feedback from all authors.

## Competing interests

The hydrogen induced cyclodehydrogenation is based on patent applications "A method for producing graphene nanostructures" (R.Z., S.G.), PCT/EP2023/060258, US 18/855,341, "Method of producing graphene nanoflakes" (R.Z., S.G.) P.440982, "Method of producing graphene nanoribbons" (R.Z., S.G.), P.440983. Apart from this, the authors declare no competing interests.
