## [Transparent Peer Review file · Nature Communications]

Cyclodehydrogenation of molecular nanographene precursors catalyzed by atomic hydrogen

Corresponding Author: Dr Szymon Godlewski

Version 0:

Reviewer comments:

Reviewer #1

(Remarks to the Author)

The manuscript describes a method of on-surface synthesis using atomic hydrogen (H^*) to catalyze cyclodehydrogenation (CDH) reactions that create carbon-carbon bonds within polycyclic aromatic hydrocarbon (PAH) precursor molecules, producing nanographenes (whose structure is determined by the precursor). The presence of H^* results in C-C bonds forming at significantly lower temperatures than for strictly thermally-induced CDH. A benefit of lower temperatures is the reduction or elimination of precursor desorption from the surface. Furthermore, the proposed reaction pathway is ostensibly independent of the substrate (normally a noble metal), so it would be applicable to semiconducting and insulating substrates as well. Although the efficiency appears to be lower than on noble metals, the authors show that indeed, H^* -promoted CDH occurs on noncatalytic semiconducting and insulating surfaces. This is the most significant aspect of the work. I recommend it for publication.

Earlier work (manuscript Ref. 39) showed a similar catalytic route for dehydrogenative polymerization and indicated its potential applicability to insulating surfaces, but Ref. 39 did not demonstrate H^* -promoted C-C bonding on an insulator and did not apply it to the important case of CDH (used to convert surface-bound polyphenylenes to nanographenes, nanoribbons, and related molecules). Others have recognized the importance of atomic hydrogen in CDH (see, e.g., Zhong et al., *J. Am. Chem. Soc.* 2024, 146, 3, 1849–1859), but without a controlled source of H^* . The submitted work demonstrates that introduction of a very modest partial pressure of H_2 ($1E-7$ mbar) is sufficient to promote CDH; this is at least 1000x smaller than required for the intermolecular C-C bonding demonstrated in Ref. 39 (at somewhat lower temperature).

The manuscript first presents evidence for H^* -catalytic action for molecules on Au(111), where STM/AFM imaging is most easily accomplished. These results clearly establish that threshold temperature for the CDH reaction is lowered by H^* . STM data from the semiconductor surfaces—which are less likely to provide their own catalytic contribution—is also convincing. Finally, the authors demonstrate H^* -induced CDH on silicon dioxide via TOF-SIMS measurements. Here the contrast between H^* -exposed and unexposed samples is subtle, but in the end, I found it convincing. The authors' experimental conclusions are established by the data presented and their proposed reaction pathway appears plausible, with support from their theoretical analysis.

This and prior work show that the influence of atomic hydrogen in on-surface synthesis can be inhibit (e.g., Ref. 36 and Kong et al., *Journal of the American Chemical Society* 2017 139 (10), 3669-3675) or facilitate (Ref. 39 and present manuscript) on-surface C-C reactions in different circumstances. Further refinement of the techniques, especially on insulating substrates as shown in the manuscript, may ultimately bring bottom-up synthesis to the device scale.

That said, some minor issues, which are OPTIONAL for the authors to address:

- 1) In Fig. 4h, blue arrows point to the molecular precursors of interest, yet the color scale saturates over every one of the precursors. Why wash out the items of interest?
- 2) In Fig. 5, perhaps tick-marks could be limited to integers on m/z scales?
- 3) Line 280: The sentence beginning in this line is slightly ambiguous. A minor change might make it clearer (provided I've interpreted correctly): "In the experiments with Si/SiO₂ we took advantage of the robustness of both precursors 6 and nanoflakes 3 and transported the samples from the preparation chamber to the SIMS setup in ambient conditions. Controlled experiments with the sample transported entirely in UHV showed comparable results."
- 4) Line 348: Hz/min is not a very useful unit of flux for someone trying to reproduce the experiments, since it depends on the

QCM characteristics and the geometry (e.g., is the QCM placed at the sample position?). It would be more useful to state the flux in physical units (e.g., molecules/cm²/sec) at the sample position.

Reviewer #2

(Remarks to the Author)

The manuscript by Zuzak and coauthors reports on the use of atomic hydrogen for cyclodehydrogenation. The work is very well executed, and I find the results well described and (in parts) well justified. However, Nature Comms doesn't appear to be the adequate vehicle for dissemination. Whilst the results are of high quality, and the work represents (on some aspects) a tour-de-force, I question the claim of novelty. The results reported here appear to be an excellent (but predictable) continuation of previous research by the same group of excellent researchers. As such, the reporting hinges on an incremental progress rather than a step change. This really constrains the novelty.

(1) The authors state that the novelty of their work resides in the application of active H towards cyclodehydrogenation of selected graphene precursors on ANY arbitrary substrate. The title does not do justice to the content of the manuscript. On face value, the title "Cyclodehydrogenation catalyzed by atomic hydrogen" does not convey any new research. The authors themselves have reported on the topic of cyclodehydrogenation catalyzed by atomic hydrogen (see their ref 32)

(2) The cyclodehydrogenation towards 2D PAHs of GNRs (thermally aided, photon assisted, or via chemical coupling) on semiconducting and insulating surfaces has been reported by the authors themselves (in their previous excellent works such as refs 29, 30) and other groups.

(3) A plausible mechanism by which H attack aids the cyclodehydrogenation reaction was (in part) reported by others (which the authors correctly reference to, see 39)

In the present manuscript, the authors piece together (1) the judicious use of active H assisted cyclodehydrogenation, and (2) apply this to semiconducting and insulating supports, thereby extending a smart synthesis recipe (operating at remarkably low temperatures) to any arbitrary substrate. The authors provide a thorough demonstration of this incremental piece of research.

A few addition queries and comments:

- The authors report up to 5-10% cracking efficiency (Methods section). Since the bulk of the work centers on the ability to deliver active hydrogen atoms, I would encourage the authors to provide more substance to their claim. 5-10% seem excessive. Most likely just 1-2%, see Jpn. J. Appl. Phys. 34 L1379).

- Fig 2, the caption says "... blue arrows indicate rarely observed incompletely planarized DBBA units". This is almost impossible to see in the 3D rendered display in 2a. I also question, here and elsewhere, the need to report shiny 3d rendered topographies (more often than not, the rendering distorts the scientific message). I recommend reporting fig 2a in a 2D perspective with perhaps an inset zooming in on an example of incompletely cyclodehydrogenated polymer. The authors should indicate the origin/identification of the many small dots observed in fig 2a alongside the GNRs (coincidentally not observed in Fig S3c). Most likely these can no longer be attributed to Br following H exposure?

- Figs 2 and 3: The effect of atomic H exposure on metal-supported GNRs has been reported by several groups (e.g. 38), with indications of alterations to the aromatic/planar/sp² network. The authors should indicate why their GNRs and nanographenes seem to be immune to these alterations. Is this due to methodology differences in sample preparation?

- Line 131: "... are shown in Fig. 3."

- Line 132: for sake of clarity, the authors should write "... the molecule HOMO-LUMO gap visualized...." and provide a reference to some form of STS data from their previous work (or include such an STS spectrum in S1).

- Lines 155 and 157: the use of "fjord" to describe the region of H attack is suboptimal.

- Fig 4. Labels on insets are unreadable (too small). Again, I question the inclusion of the slick 3D rendering in panel C. Panels D and E conveys the message adequately, so C can be removed.

- Ample reports exist on the exposure of atomic H onto the surfaces of titania with the emergence of hydroxyls. Why aren't these observed in any significant concentration in Figs 4a,b,g? Might be some evidence in 4f? The authors should comment on the effect of H exposure on their substrates.

- As the insets of figs 4a and 4g demonstrate, planarization of their precursors is accompanied by an electronic height decrease. As amply reported for metal supports, the 2D-PAHs exhibit an apparent height of about 0.25-0.27 nm (see 4a), whereas on TiO₂ supports, the height is about 0.35 nm. The height decrease upon planarization is marginal on TiO₂ (perhaps about 0.1 nm) in comparison to metal supports (almost 0.25 nm). Nb: I estimate these values directly from fig 4. Why is there such a difference? Can the authors provide most substance to convince the readership the planarization on the reduced oxide is complete? What is the bias-dependence on apparent height? I expect it to be negligible on Au, but significant on TiO₂ or NaCl/metal.

- Sentence on lines 240-242: poorly formulated.

- Direct visualization of the planar reaction products on Si/SiO₂ would be a massive bonus to complement the ToF-SIMS data. Mindful of the challenging nature of such measurements, the authors should comment on attempts?

Reviewer #3

(Remarks to the Author)

Zuzak and coauthors reported the high effective cyclodehydrogenation catalyzed by atomic hydrogen in a manner independent of the substrates. With the extensive STM/nc-AFM and ToF-SIMS experiments on metallic Au(111), semiconducting TiO₂, Ge:H, as well as insulating Si/SiO₂ and thin NaCl layers, the authors illustrated the catalytic role of atomic hydrogen for the cyclodehydrogenative planarization of graphene-like structures. Previously extensive study of the

atomic-hydrogen-protocol in the on-surface synthesis field has demonstrated the unique role of tailoring reactions on metal surfaces. In this work, the research team further extended its application in the planarization reaction of graphene derivatives on flat surfaces, and in particular, on non-metal substrates. In total this is an interesting study which certainly contributes to the on-surface synthesis field by expanding the atomic-hydrogen-protocol to the precise synthesis of functional nanostructures on non-metal substrates.

The idea is original and the manuscript is well written. The methodology is described detail enough to be reproduced. The experimental work is systematic and abundant, while the plausible reaction mechanism might need to be advised. Therefore I suggest the authors address several concerns as listed below before its acceptance.

1) The designed experiments demonstrate well the catalytic effect of atomic hydrogen by comparing the cyclodehydrogenation temperatures with/without externally dosed atomic hydrogen treatment. But the attempts to explain the reaction mechanism are somewhat deficient, especially the initial step of atomic H addition to the peripheral C atom. In Fig. S4a, the authors select 10 as the first hydrogenated intermediate of model compound 9 with hydrogen added manually to the peripheral C atom forming a π -radical on the fjord region. The lack of discussion on the alternative hydrogen addition choice significantly reduces its rigor and credibility.

2) The authors performed gas-phase DFT calculations to illustrate the catalytic action of atomic hydrogen regardless of the substrate type. And the resulted Gibbs free-energy profile shows relatively small barrier value (0.56 eV), which does not match the energy corresponding to the actual experimental temperature (220°C). It is better to add discussion o the mismatching or illustrate the effect of omitted substrate.

3) While reviewing the examples of on-surface synthesis on nonmetallic substrates in the introduction, it would be inappropriate to say "relatively scarce", assessing it to be inefficient or non-universal.

4) In addition, in the last part of the manuscript, it would be good if there is a conclusive summary of the work besides the ascensive perspectives.

Version 1:

Reviewer comments:

Reviewer #1

(Remarks to the Author)

I am satisfied with the authors' responses to all reviewers. I recommend publication, with one small edit:

Line 123: "formed thorough" should be "formed through"

Reviewer #2

(Remarks to the Author)

The authors have done a diligent work at considering the comments made by the three referees. The changes brought to the manuscripts as a result of the suggestions for change have improved the manuscript significantly. I've also carefully considered the authors rebuttal of my concerns in relation to the incremental nature of their research. I admit, as the authors correctly pointed out, to my confusion in relation to their previous work and interpretation of it. I apologize for the oversight and misunderstanding. Under this new light, and paralleled with the assessment made by the other referees, the results presented by Zuzak and coworkers merit dissemination via Nature Communication.

Reviewer #3

(Remarks to the Author)

The authors have carefully addressed my comments and have considerably improved their manuscript. I feel the revised MS are convincing enough thus to suggest the acceptance for publication.

Response to Reviewer Comments:

Reviewer 1:

Comment:

The manuscript describes a method of on-surface synthesis using atomic hydrogen (H^*) to catalyze cyclodehydrogenation (CDH) reactions that create carbon-carbon bonds within polycyclic aromatic hydrocarbon (PAH) precursor molecules, producing nanographenes (whose structure is determined by the precursor). The presence of H^* results in C-C bonds forming at significantly lower temperatures than for strictly thermally-induced CDH. A benefit of lower temperatures is the reduction or elimination of precursor desorption from the surface. Furthermore, the proposed reaction pathway is ostensibly independent of the substrate (normally a noble metal), so it would be applicable to semiconducting and insulating substrates as well. Although the efficiency appears to be lower than on noble metals, the authors show that indeed, H^* -promoted CDH occurs on noncatalytic semiconducting and insulating surfaces. This is the most significant aspect of the work. I recommend it for publication.

Earlier work (manuscript Ref. 39) showed a similar catalytic route for dehydrogenative polymerization and indicated its potential applicability to insulating surfaces, but Ref. 39 did not demonstrate H^* -promoted C-C bonding on an insulator and did not apply it to the important case of CDH (used to convert surface-bound polyphenylenes to nanographenes, nanoribbons, and related molecules). Others have recognized the importance of atomic hydrogen in CDH (see, e.g., Zhong et al., J. Am. Chem. Soc. 2024, 146, 3, 1849–1859), but without a controlled source of H^* . The submitted work demonstrates that introduction of a very modest partial pressure of H_2 ($1E-7$ mbar) is sufficient to promote CDH; this is at least 1000x smaller than required for the intermolecular C-C bonding demonstrated in Ref. 39 (at somewhat lower temperature).

The manuscript first presents evidence for H^* -catalytic action for molecules on Au(111), where STM/AFM imaging is most easily accomplished. These results clearly establish that threshold temperature for the CDH reaction is lowered by H^* . STM data from the semiconductor surfaces---which are less likely to provide their own catalytic contribution---is also convincing. Finally, the authors demonstrate H^* -induced CDH on silicon dioxide via TOF-SIMS measurements. Here the contrast between H^* -exposed and unexposed samples is subtle, but in the end, I found it convincing. The authors' experimental conclusions are established by the data presented and their proposed reaction pathway appears plausible, with support from their theoretical analysis.

This and prior work show that the influence of atomic hydrogen in on-surface synthesis can be inhibit (e.g., Ref. 36 and Kong et al., Journal of the American Chemical Society 2017 139 (10), 3669-3675) or facilitate (Ref. 39 and present manuscript) on-surface C-C reactions in different circumstances. Further refinement of the techniques, especially on insulating substrates as shown in the manuscript, may ultimately bring bottom-up synthesis to the device scale.

Answer:

We thank for the positive assessment of our work and constructive comments. We are also very grateful for the detailed analysis of the novelty presented in our manuscript and the overall

evaluation of achievements in the context of on-surface synthesis development and prospects for the bottom-up approach device fabrication.

In the revised manuscript we have added the reference to the recent manuscript by Zhong et al. (J. Am. Chem. Soc. 2024, 146, 3, 1849–1859) as ref 29, which describes influence of atomic hydrogen, (however as mentioned by the reviewer without a controlled source), on a different reaction compared to cyclodehydrogenation (CDH) analysed in our manuscript, namely the H shift and elimination reactions.

Comment:

- 1) In Fig. 4h, blue arrows point to the molecular precursors of interest, yet the color scale saturates over every one of the precursors. Why wash out the items of interest?

Answer:

Following the comment we have modified the color scale of Figure 4h.

Comment:

- 2) In Fig. 5, perhaps tick-marks could be limited to integers on m/z scales?

Answer:

Following the recommendation we have modified the tick-marks to improve the readability.

Comment:

- 3) Line 280: The sentence beginning in this line is slightly ambiguous. A minor change might make it clearer (provided I've interpreted correctly): "In the experiments with Si/SiO₂ we took advantage of the robustness of both precursors 6 and nanoflakes 3 and transported the samples from the preparation chamber to the SIMS setup in ambient conditions. Controlled experiments with the sample transported entirely in UHV showed comparable results."

Answer:

Following the comment we have modified the text accordingly.

Comment:

- 4) Line 348: Hz/min is not a very useful unit of flux for someone trying to reproduce the experiments, since it depends on the QCM characteristics and the geometry (e.g., is the QCM placed at the sample position?). It would be more useful to state the flux in physical units (e.g., molecules/cm²/sec) at the sample position.

Answer:

Based on the recommendation we have calculated the molecular flux reaching the sample and supplemented the data with the required value.

Reviewer 2:

Comment:

The manuscript by Zuzak and coauthors reports on the use of atomic hydrogen for cyclodehydrogenation. The work is very well executed, and I find the results well described and (in parts) well justified. However, Nature Comms doesn't appear to be the adequate vehicle for dissemination. Whilst the results are of high quality, and the work represents (on some aspects) a tour-de-force, I question the claim of novelty. The results reported here appear to be an excellent (but predictable) continuation of previous research by the same group of excellent researchers. As such, the reporting hinges on an incremental progress rather than a step change. This really constrains the novelty.

(1) The authors state that the novelty of their work resides in the application of active H towards cyclodehydrogenation of selected graphene precursors on ANY arbitrary substrate. The title does not do justice to the content of the manuscript. On face value, the title "Cyclodehydrogenation catalyzed by atomic hydrogen" does not convey any new research. The authors themselves have reported on the topic of cyclodehydrogenation catalyzed by atomic hydrogen (see their ref 32)

Answer:

There must be an important confusion here, since ref 32 of the original submission (revised manuscript ref. 33) does not refer to cyclodehydrogenation reactions at all. In fact, we would like to emphasize that to the best of our knowledge the cyclodehydrogenation catalyzed by atomic hydrogen has never been reported. We note that the ref 32 describes application of atomic hydrogen (only on metals) to: (1) act as a cleaning tool, which allows removal of halogen residues from the surface after Ullmann couplings/polymerization, (2) quench organometallic intermediates, and (3) act as a reagent for debromination or desulfurization of adsorbed species. All aforementioned areas are completely unrelated to planarization/cyclodehydrogenation reactions being the subject of our present study. Therefore, we consider such reviewer statements to be unfounded and we kindly ask the reviewer to reconsider his/her decision. We refer here to very detailed opinion of the first reviewer, who has emphasized the most important aspect of our work, which is the ability to induce planarization into graphene – based species in noncatalytic surfaces - "Furthermore, the proposed reaction pathway is ostensibly independent of the substrate (normally a noble metal), so it would be applicable to semiconducting and insulating substrates as well. Although the efficiency appears to be lower than on noble metals, the authors show that indeed, H-promoted CDH occurs on noncatalytic semiconducting and insulating surfaces. This is the most significant aspect of the work".*

To make the title of our manuscript more precisely describing our findings we have modified it to the following form: "Cyclodehydrogenation of nanographene molecular precursors catalyzed by atomic hydrogen".

Comment:

2) The cyclodehydrogenation towards 2D PAHs of GNRs (thermally aided, photon assisted, or via chemical coupling) on semiconducting and insulating surfaces has been reported by the authors themselves (in their previous excellent works such as refs 29, 30) and other groups

Answer:

We humbly disagree with this statement. To the best of our knowledge the controlled cyclodehydrogenation into planar nanoflakes has been reported only on TiO₂(110) by some of us [ref 29 in original submission, ACS Nano 17, 2580–2587 (2023)] and dehydrogenation into nanodomains also on TiO₂(110) by Carlos Sánchez-Sánchez et al. in Nanoscale, 2013,5, 11058-11065 (which we have added as a reference in the revised manuscript). Other examples leading to atomically precise planar nanoflakes/GNRs [refs 30, 31 of the original submission, i.e. Science 363, 57–60 (2019) and Science 369, 571–575 (2020)] are based on the use of fluoride-substituted precursors and application of a different reaction (i.e. cyclodehydrodefluorization reaction), which has been reported only on TiO₂(011) so far. Therefore, we could describe the aforementioned examples as greatly interesting, but limited to very specific faces of a specific substrate. Finally, according to the best of our knowledge the planarization reaction into atomically precise graphene nanoflakes has not been reported at all on insulators. Therefore, we believe that the demonstration of a planarization reaction on a large range of substrates, together with the catalytic role of externally dosed atomic hydrogen in on-surface synthesis cannot be considered as incremental work. The novelty of our approach has also been emphasized by reviewers 1 and 3.

Comment:

(3) A plausible mechanism by which H attack aids the cyclodehydrogenation reaction was (in part) reported by others (which the authors correctly reference to, see 39)

Answer:

We politely disagree with this statement. In ref 39 [original submission, i.e. J. Am. Chem. Soc. 141, 3550-3557 (2019)] the uncontrolled intermolecular coupling on Au(111) induced by atomic hydrogen is reported, while in our manuscript we describe a well-defined hydrogen catalyzed intramolecular cyclodehydrogenation.

Comment:

In the present manuscript, the authors piece together (1) the judicious use of active H assisted cyclodehydrogenation, and (2) apply this to semiconducting and insulating supports, thereby extending a smart synthesis recipe (operating at remarkably low temperatures) to any arbitrary substrate. The authors provide a thorough demonstration of this incremental piece of research.

Answer:

We emphasize once again that the application of atomic hydrogen to initiate cyclodehydrogenation reactions has not been reported so far. Furthermore we could not agree with the statement that performing efficient cyclodehydrogenation on insulating and semiconducting substrates is an incremental piece of research, as the precise synthesis of GNRs/nanoflakes on such technologically relevant substrates has been a long-standing challenge and the development of the approach which enables overcoming limitations of metal catalyzed on-surface reactions provides a step forward in the precise generation of graphene – based units, as emphasized by reviewers 1 and 3.

Comment:

- The authors report up to 5-10% cracking efficiency (Methods section). Since the bulk of the work centers on the ability to deliver active hydrogen atoms, I would encourage the authors to

provide more substance to their claim. 5-10% seem excessive. Most likely just 1-2%, see Jpn. J. Appl. Phys. 34 L1379).

Answer:

We are grateful for the comment. The precise efficiency of the setup is difficult to estimate. In the cited work (Jpn. J. Appl. Phys. 34 L1379) the authors have estimated the efficiency of hydrogen cracking in their system to vary between 1,5% and saturation value of 3% depending on the filament temperature. However, they show also for comparison the data obtained for a commercially available system (also with hot filament, Figure 5), which shows exponential increase of efficiency with temperature, i.e. the efficiency increases from approximately 2% at 1600 °C, through 5% at 2000 °C up to 13% at 2300 °C. The authors explain the difference by the optimized design of the commercial unit. Taking into account the difficulty with the precise estimation of the cracking efficiency and following the reviewer comment we have modified the description leaving only the upper estimate limit, which we believe does not exceed 10% and rephrased the description to sound: “The estimated efficiency of molecular hydrogen cracking is below 10%”

Comment:

- Fig 2, the caption says “... blue arrows indicate rarely observed incompletely planarized DBBA units”. This is almost impossible to see in the 3D rendered display in 2a. I also question, here and elsewhere, the need to report shiny 3d rendered topographies (more often than not, the rendering distorts the scientific message). I recommend reporting fig 2a in a 2D perspective with perhaps an inset zooming in on an example of incompletely cyclodehydrogenated polymer. The authors should indicate the origin/identification of the many small dots observed in fig 2a alongside the GNRs (coincidentally not observed in Fig S3c). Most likely these can no longer be attributed to Br following H exposure?

Answer:

Following the recommendation, and in order to improve the readability, we have modified the graphics and replaced 3D renders with standard 2D ones.

The reviewer is right that the small dots in Fig.2a are not bromine atoms, they are originating from DBBA molecules, which were planarized by atomic hydrogen treatment and did not undergo polymerization reaction (before planarization). We note here that these species are predominantly located within the “elbows” of the Au(111) reconstruction pattern, which are the most active sites on the surface. These species are also noticeable in Figure S3c between nanoribbons (see below). For clarity we have added the explanation to the figure caption.

Comment:

- Figs 2 and 3: The effect of atomic H exposure on metal-supported GNRs has been reported by several groups (e.g. 38), with indications of alterations to the aromatic/planar/sp² network. The authors should indicate why their GNRs and nanographenes seem to be immune to these alterations. Is this due to methodology differences in sample preparation?

Answer:

As clearly demonstrated by our nc-AFM measurements we have not observed transformation from sp^2 into sp^3 carbon atoms located at armchair edges, whereas a fraction of more reactive zigzag edge located carbon atoms is observed to transform in $-CH_2-$ units in the temperature range of 160-220 °C with the pressure in the low 10^{-7} mbar range. We note here that in ref. 38 (ACS Nano 16, 10281-10291 (2022)) the authors were applying atomic hydrogen at room temperature and at higher pressure than in our experiments. We note here that also the experiments reported in ref 39 (J. Am. Chem. Soc. 141, 3550-3557 (2019)) leading to hydrogenation of nanoflake edges and uncontrolled intermolecular fusion were performed with higher hydrogen pressure. We conclude that both factors, i.e. the temperature and the pressure range during hydrogen treatment influences the course of the reaction resulting in different reaction outcomes obtained in our experiments and in the research reported in refs 38-39 of the original submission (i.e. ACS Nano 16, 10281-10291 (2022) and J. Am. Chem. Soc. 141, 3550-3557 (2019)).

Comment:

- Line 131: "... are shown in Fig. 3."

- Line 132: for sake of clarity, the authors should write "... the molecule HOMO-LUMO gap visualized..." and provide a reference to some form of STS data from their previous work (or include such an STS spectrum in SI).

- Lines 155 and 157: the use of "fjord" to describe the region of H attack is suboptimal.

- Fig 4. Labels on insets are unreadable (too small). Again, I question the inclusion of the slick 3D rendering in panel C. Panels D and E conveys the message adequately, so C can be removed.

Answer:

Following the recommendations we have modified the manuscript accordingly. We note here that the term "fjord" is used in chemistry for the described part of the molecule, therefore we have decided to keep the term "fjord" as a supplement in brackets.

We have also replaced the 3D image by the standard 2D one. We believe that its inclusion is justified, because in contrast to panel d presenting mobile GNR, panel c shows GNR in a stable configuration. We have also enlarged panels a-e and increased the size of the insets to make the figure more readable.

Comment:

- Ample reports exist on the exposure of atomic H onto the surfaces of titania with the emergence of hydroxyls. Why aren't these observed in any significant concentration in Figs 4a,b,g? Might be some evidence in 4f? The authors should comment of the effect of H exposure on their substrates.

Answer:

We thank for the remark. In our previous experiments with TiO_2 focused on the influence of the surface hydroxyls on the on-surface polymerization efficiency (Chem. Commun., 2015,51, 11276-11279) we have extensively studied the effect of atomic hydrogen on the (011) surface of titania. In these studies we have attempted preparation of the hydroxylated surface with a

controlled density of surface hydroxyls. Based on the experiments we have concluded that the most efficient surface hydroxylation is achieved when the titania surface is exposed to the source of atomic hydrogen at room temperature (compared to elevated temperature). These findings are in agreement with our current studies of both $\text{TiO}_2(110)$ and $\text{TiO}_2(011)$ surfaces, for which we do not observe increase of the surface hydroxyls density when the samples are kept at elevated temperature during treatment with atomic hydrogen. This is already visible in Figure 4f,g for $\text{TiO}_2(011)$ as the reviewer rightly notes. To illustrate the issue for the $\text{TiO}_2(110)$ surface we attach below additional STM image of the surface with planarized nanographenes (Fig. A1).

Fig. A1. STM image of nanographenes **3** on $\text{TiO}_2(110)-(1 \times 1)$, bias voltage: +1.5 V, tunneling current: 20 pA.

Moreover, we note here that heating of titania samples above 200 °C may lead to the decrease density of surface hydroxyls. This observation is also in agreement with previous studies (e.g. *Surface Science* 598 (2005) 226–245).

Following the reviewer remark we discuss further the effect of H exposure to other surfaces. We do not observe any influence of atomic hydrogen dosing on $\text{Au}(111)$. The NaCl surface is also unaltered, the detailed analysis (atomic resolution imaging) has been obtained for the bulk NaCl with the application of the nc-AFM technique (already presented in the original submission as Figure S7 (S8 in the revised version) in Supplementary Material). The SiO_2 surface has not been examined in our experiments with scanning probe techniques, although we refer here to ref. 48 of the original submission (*Phys. B: Condens. Matter.* 533, 5-11 (2018)), which reports on the positive influence of atomic hydrogen on the surface quality. This indicates that the hydrogen treatment of the amorphous SiO_2 surface would not deteriorate its quality. Finally, regarding the hydrogen protected $\text{Ge}(001):\text{H}$ surface, we note that the hydrogen dosing procedure aimed at cyclodehydrogenation of molecular precursors is performed within almost

identical conditions to those applied in order to achieve high quality passivation by hydrogen. This procedure has been established in our previous experiments (e.g. Phys. Rev. B 86, 125307 (2012)) and we could conclude that the hydrogenation performed within the applied temperature window (160-220 °C) and in the 10^{-7} mbar pressure range does not lead to surface damage (e.g. etching, which could occur at different conditions). We hope that the explanation sufficiently explains the issue of the impact of atomic hydrogen on the applied surfaces.

Comment:

- As the insets of figs 4a and 4g demonstrate, planarization of their precursors is accompanied by an electronic height decrease. As amply reported for metal supports, the 2D-PAHs exhibit an apparent height of about 0.25-0.27 nm (see 4a), whereas on TiO₂ supports, the height is about 0.35 nm. The height decrease upon planarization is marginal on TiO₂ (perhaps about 0.1 nm) in comparison to metal supports (almost 0.25 nm). Nb: I estimate these values directly from fig 4. Why is there such a difference? Can the authors provide most substance to convince the readership the planarization on the reduced oxide is complete? What is the bias-dependence on apparent height? I expect it to be negligible on Au, but significant on TiO₂ or NaCl/metal.

Answer:

*The reviewer is right that the apparent height of the planar nanographenes on TiO₂ strongly depends on the applied bias voltage. Below we demonstrate the profiles recorded over nanographenes **3** on TiO₂(110) for which we have managed to acquire the images in the wider range of voltages. We note here that the STM imaging of filled states is usually unstable on TiO₂ surfaces. The profiles clearly show that the apparent height for positive voltages (empty state imaging) is quite similar both for voltages corresponding to molecule band gap (Figure A2b, blue profile in Figure A2d), as well as for voltages exceeding the energy position of LUMO state (Figure A2b, red profile in Figure A2d). The situation is different for filled state imaging at moderate voltages as exemplified in Figure A2c (green profile in Figure A2d).*

*Recording images (Figure 3g-j of the main text), which resemble HOMO (Figure A2c) and LUMO (Figure A2a) states of the nanographene **3** demonstrates the successful planarization on the TiO₂(110) surface.*

Figure A2. Nanographenes **3** on $\text{TiO}_2(110)$, high resolution STM images acquired at +2.2V (a), +1.5V (b) and -1.5V (c) together with cross-sections (d).

The recorded apparent height of nanoflakes **3** for two different bias voltages was quite comparable as shown in Figure A3, although we note here that due to the mobility of the molecules we have not been able to acquire high resolution STM images for other voltage values.

Figure A3. STM images (a, b) of nanoflake **3** on NaCl/Cu together with the image cross-sections (c).

The reviewer asks about the data presented for the $\text{TiO}_2(011)$ surface presented in Figure 4f,g of the main text, which indeed show limited height difference between the precursor **6** and

nanoflake **3**. In case of the $\text{TiO}_2(011)$ surface we have not been able to perform stable measurements in the filled state regime. However, we have performed the STM measurements of nanoflakes **6** for different positive voltages and we discuss the issue in details below.

Figures A4a and A4b show images recorded over precursors **6** (panel a) and nanoflakes **3** (panel b) at different voltages following the data shown in the main text Figure 4f,g. We note here that the image of precursor **6** exhibits clearly 3 lobes corresponding to the three areas of the steric hindrance leading to the highly non-planar conformation of the precursor (similarly as Fig.4f). In contrast Figure A4b (and Figure 4g) show nanographene **3** with intramolecular contrast, which combines the LUMO state distribution (the image is recorded at +2.3V, which is above the LUMO state energy position, see Figure A5 below) and the anisotropy of the surface (we have already observed the influence of the anisotropic substrate surface on the molecule appearance e.g. on $\text{Ge}(001):\text{H}$ – ACS Nano 2013, 7, 10105). Our intention was to demonstrate the intramolecular contrast resembling LUMO state of nanographene **3** and we believe that the characteristic triangular shape is well visible in the revised version of Figure 4g both for the molecule on the terrace as well as for those attached to the step edge. However, recording the images of precursors **6** and nanoflakes **3** at different voltages leads to moderate difference in their apparent height (inset in Figure 4g, blue and green profiles in Figure A4c). Therefore in order to shed more light and provide additional evidence for the planarization we present below additional analysis. In Figure A4c (yellow line) we present scan profiles across nanoflakes **3** on $\text{TiO}_2(011)$ recorded at +1.5V (corresponding to the gap of the molecule **3**) recorded in the image shown in Figure A4e. We note here that the scan profiles drawn in Figure A4c in green (precursor **6**) and in yellow (nanoflake **3**) show significant height difference. Moreover the image recorded at +1.5V (Figure A4e and S5b from Supplementary Material) indicates that the molecules exhibit a planar shape resembling the geometrical structure of nanographene **3** (which is expected for voltages corresponding to the molecule gap). Additionally, we show that the image of **3** on $\text{TiO}_2(011)$ as well as the height profile is comparable with the data recorded for nanoflake **3** on $\text{TiO}_2(110)$ (Figure A4d, red profile in Figure A4c). We believe that this data sufficiently demonstrates the successful planarization of **6** into **3** also on $\text{TiO}_2(011)$. We note here that we record the STM images at +1.5V for high coverage to increase the stability of the molecules during imaging and to avoid characterization only of those molecules, which are immobilized by defects. For the sake of completeness we have decided to add the newly prepared Figure A4 also as a new figure in Supplementary Material as new Fig. S6.

Figure A4. STM images of precursors **6** and nanographenes **3**, (a) **6** on $\text{TiO}_2(011)$, (b) and (e) **3** on $\text{TiO}_2(011)$, (c) image cross-sections from panels a,b,d,e, (d) **3** on $\text{TiO}_2(110)$.

Figure A5. STS data recorded over nanoflake **3** on $\text{TiO}_2(011)$ -(2x1).

Comment:

- Sentence on lines 240-242: poorly formulated.

Answer:

Following the comment we have modified the description to make it more straightforward:

Additional peaks recorded at masses starting from 816 up to 827 Da could be attributed to the precursors in various states of dehydrogenation induced by ToF-SIMS measurements, as already reported for precursors on Au(111) [51]. The tail extended exactly to the fully dehydrogenated nanographene **3** in full accordance with the previous report on a metallic Au(111) substrate [51].

Comment:

- Direct visualization of the planar reaction products on Si/SiO₂ would be a massive bonus to complement the ToF-SIMS data. Mindful of the challenging nature of such measurements, the authors should comment on attempts?

Answer:

*We thank for the comment. Direct visualization of the transformation of **6** into **3** would require challenging high resolution AFM imaging. Such experiment would be even more difficult to perform on the SiO₂ surface, because to the best of our knowledge submolecular resolved imaging of molecular species has not been reported for such surfaces so far. Furthermore, we need to note that we have taken up the challenge to image the transformation of **6** into **3** with high resolution AFM on bulk NaCl, which is much better known in the high resolution AFM community. However, to our regret, we have not been able to record sufficiently resolved*

*images to demonstrate the transformation. In addition, it is worth to emphasize that high resolution AFM imaging on bulk insulators is relatively slow and very time – consuming. Therefore, it would be a huge challenge to record convincing images with sufficient statistics. On the other hand, we believe that the application of ToF-SIMS technique is an interesting alternative providing the averaged information, which is therefore not based on single molecule examples. Additionally, we note here that we have combined STM and ToF-SIMS measurements for the very same samples (experiments with **6** and **3** on Au(111)) and both ToF-SIMS and STM measurements were convincingly showing presence of **6** before transformation and **3** after planarization. Therefore, taking into account the fact that the experiment would require design and implementation of the high-resolution AFM imaging on an amorphous surface without the know ability to functionalize the tip apex we believe that such enormous experimental effort is beyond the scope of our current manuscript and as emphasized by reviewer 1 our ToF-SIMS data provide convincing confirmation of the transformation of **6** into **3** on bulk insulator.*

Reviewer 3:

Comment:

Zuzak and coauthors reported the high effective cyclodehydrogenation catalyzed by atomic hydrogen in a manner independent of the substrates. With the extensive STM/nc-AFM and ToF-SIMS experiments on metallic Au(111), semiconducting TiO₂, Ge:H, as well as insulating Si/SiO₂ and thin NaCl layers, the authors illustrated the catalytic role of atomic hydrogen for the cyclodehydrogenative planarization of graphene-like structures. Previously extensive study of the atomic-hydrogen-protocol in the on-surface synthesis field has demonstrated the unique role of tailoring reactions on metal surfaces. In this work, the research team further extended its application in the planarization reaction of graphene derivatives on flat surfaces, and in particular, on non-metal substrates. In total this is an interesting study which certainly contributes to the on-surface synthesis field by expanding the atomic-hydrogen-protocol to the precise synthesis of functional nanostructures on non-metal substrates.

The idea is original and the manuscript is well written. The methodology is described detail enough to be reproduced. The experimental work is systematic and abundant, while the plausible reaction mechanism might need to be advised. Therefore I suggest the authors address several concerns as listed below before its acceptance.

Answer:

We are grateful for the positive assessment of our manuscript.

Comment:

1) The designed experiments demonstrate well the catalytic effect of atomic hydrogen by comparing the cyclodehydrogenation temperatures with/without externally dosed atomic hydrogen treatment. But the attempts to explain the reaction mechanism are somewhat deficient, especially the initial step of atomic H addition to the peripheral C atom. In Fig. S4a, the authors select 10 as the first hydrogenated intermediate of model compound 9 with hydrogen added manually to the peripheral C atom forming a π -radical on the fjord region. The lack of discussion on the alternative hydrogen addition choice significantly reduces its rigor and credibility.

Answer:

We appreciate the referee's thorough review of our manuscript and for highlighting this important aspect. Of course the H addition can take place in other molecular regions to form other radicals in vicinal positions. However, only a π -radical in the fjord position would evolve to form a C-C bond. In other situations, the H addition would be reversible in the presence of atomic H, that is, would lead to C-H cleavage to recover a closed-shell structure (similar to **13** to **14** transformation in Figure S4).

Following the reviewer comment we present below the additional data, which corroborated the data presented in the manuscript. We initially analyzed hydrogen addition to the carbon atom that leads to the activation of the neighboring carbon, which is crucial for initiating the planarization mechanism with the initial C—C cyclization. Nevertheless, to address this concern comprehensively, we have conducted additional CI-NEB calculations on the remaining non-equivalent (by symmetry) peripheral carbon atoms to compare their behavior with the one previously studied (TS1-TS7 in Fig. A6). These calculations include the determination of minimum reaction paths for hydrogen addition to the selected carbon atoms, as depicted in the figure (with TS1 being the one previously analyzed for comparison). The results indicate that hydrogen addition to peripheral carbon atoms forming C—H bonds yields adsorption energies ranging from -1.28 to -1.58 eV, with reaction barriers between 0.08 eV (the lowest value, corresponding to TS1) and 0.33 eV for TS4. For the other two peripheral carbon atoms (TS5 and TS7), the adsorption energy is approximately 1 eV higher (less favorable) compared to the rest, with barriers of around 0.25 eV. These new calculations reveal that hydrogen addition to the carbon atom critical for activating the proposed mechanism, and the only one having significant chemical implications for our purposes, is theoretically predicted to be the preferential adsorption site. Besides, there are numerous examples in the literature, including studies on coronenes and similar molecular precursors, reporting hydrogen adsorption energies for all carbon atoms within the structure with similar results [e.g. *The Astrophysical Journal* 2008, 679, 531; *Phys. Chem. Chem. Phys.* 2019, 21, 12012-12020]. Additionally, we computed the minimum energy paths for hydrogen Eley-Rideal abstraction on two representative carbon atoms (TS2 and TS5) by the same formalism and protocol to those reported in the manuscript. These processes resulted, similarly, in barrierless pathways. We believe that the additional explanation dispel the reviewer's concerns.

Figure A6. Potential energy curves for addition of hydrogen atom into all inequivalent 7 different edging C sites (TS1-TS7) of the model precursor.

Comment:

2) The authors performed gas-phase DFT calculations to illustrate the catalytic action of atomic hydrogen regardless of the substrate type. And the resulted Gibbs free-energy profile shows relatively small barrier value (0.56 eV), which does not match the energy corresponding to the actual experimental temperature (220°C). It is better to add discussion on the mismatching or illustrate the effect of omitted substrate.

Answer:

We would like to thank the referee for her/his insightful comment, which deserves some clarification. As reported, our calculations indicate that, within the theoretically predicted atomistic reaction mechanism, the limiting reaction step identified, the one requiring a higher activation temperature, is the cyclization via C—C coupling following the activation of the molecule after H addition to the peripheral C atom. This sub-reaction has yielded a computed barrier of 0.56 eV, strictly translating into a high temperature far from the operational temperature of 220°C. However, at that temperature, Boltzmann and Arrhenius formalisms predict that that barrier can be surpassed in typical times of around 10^{-4} seconds, which perfectly aligns with the observations.

Experimental evidence shows that the catalytic role of atomic hydrogen from the cracker is sufficient to initiate and complete the planarization process, regardless of the substrate's nature; the different substrates seem to play the role of confining scaffolds for the molecular precursors. This led us to elucidate the atomistic reaction mechanism in the gas-phase as a global workbench scenario. By computing the higher barrier for the cyclization sub-reaction in the gas-phase, we ensure this value as an authentic upper bound for the on-surface cases. Any influence of a substrate (beyond molecular physisorption regimes, which do not involve subtle chemistry in the molecule/substrate interaction) would decrease this value by quenching the molecular properties. The main factor from the substrates that could reduce this value to match the operational temperature better is the availability of diffusing atomic hydrogen on the surfaces (either atomic hydrogen not-captured by molecular precursors directly coming from the H-cracker at first or atomic hydrogen detached from the already planarized precursors towards the surfaces according to the mechanism). This could significantly increase the atomic hydrogen capture cross-section by pristine precursors, enhancing the planarization process in time and efficiency, which our gas-phase calculations do not account for, as detached H atoms are considered to go to infinity. On the other hand, we could conjecture some other surface effects that may play a role reducing the limiting step barrier: i) the appearance of image potentials (case of Au), yielding Coulomb-like attractive image forces, or ii) the surface-driven electrostatic slight structural distortions into the precursors inducing the appearance of intrinsic molecular dipoles enhancing the catalysis of the process, among others.

Additionally, we performed further calculations to explore another unlikely process. One might speculate that simultaneous double H addition, rather than a single H addition to one

peripheral C atom, but to both peripheral C atoms (each near the coupling C—C atoms leading to cyclization), could reduce or eliminate the energy barrier (see Fig. A7). We repeated the minimum energy path (MEP) calculation of the cyclization sub-reaction with double H addition to test this hypothesis. The initial state, four intermediate steps, and the final step of the MEP calculation showed that cyclization indeed occurs with no barrier. However, we consider this highly improbable since the likelihood of simultaneous double H addition in this manner is practically zero. Moreover, a second H addition would already be affected by changes in the molecular electronic properties induced by the first H addition.

Figure A7. Analysis of the double hydrogen addition to the precursor.

In order to clarify the discussion in the manuscript we have extended the description with the additional part:

Therefore, the aforementioned C-C bond formation is the limiting reaction step, featuring a moderate energy barrier of 0.56 eV making the process highly probable in the experimentally applied temperature range in typical times of around 10^{-4} seconds (for details see SI Fig. S4). These gas-phase calculations ensure that limiting reaction step energy barrier of 0.56 eV as an authentic upper bound for the on-surface cases. Influence of any of the substrates under study in the atomistic mechanism would just decrease that energy barrier value due to different factors like, e.g.: (i) a higher available amount of atomic hydrogen diffusing on the surface (increasing the atomic hydrogen capture cross-section by the deposited precursors, boosting the planarization process in time and efficiency), (ii) the appearance of interfacial image potentials (case of Au), yielding Coulomb-like attracting image forces, or (iii) the surface-driven electrostatic interaction leading to slight structural distortions of the precursors inducing the appearance of intrinsic molecular dipoles, thus so enhancing the catalysis of the process, just to mention a few. Besides, additional calculations reveal that, although very unlikely, a simultaneous double H addition, rather than a single H addition to one peripheral C atom, but

to both peripheral C atoms (each near the coupling C—C atoms leading to cyclization), eliminates the cyclization energy barrier.

Comment:

3) While reviewing the examples of on-surface synthesis on nonmetallic substrates in the introduction, it would be inappropriate to say “relatively scarce”, assessing it to be inefficient or non-universal.

Answer:

Following the comment we have rephrased the sentence.:

Examples of on-surface synthesis on nonmetallic substrates are rarely reported [16-28], and the planarization pathways have been limited to individual examples of surface assisted dehydrogenation [29], cyclodehydrogenation [30] or cyclodehydrodefluorination [31, 32] on very specific faces of rutile titania crystal.

Comment:

4) In addition, in the last part of the manuscript, it would be good if there is a conclusive summary of the work besides the ascensive perspectives.

Answer:

Following the recommendation we have updated the manuscript with a short summary of the most significant outcome of our research:

To sum up, we have demonstrated that the planarization of nanographene molecular precursors could be achieved through cyclodehydrogenation catalyzed by atomic hydrogen on different substrates. This constitutes a promising synthetic route to overcome the long-standing challenge to initiate molecular transformations on inactive surfaces. Therefore, the atomically precise formation of nanographenes is no longer limited to catalytically active surfaces, providing an approach which enables performing on-surface synthesis on chemically inert surfaces.